# GENERATIVE ADVERSARIAL NETS FOR MULTIPLE TEXT CORPORA

## ABSTRACT

Generative adversarial nets (GANs) have been successfully applied to the artificial generation of image data. In terms of text data, much has been done on the artificial generation of natural language from a single corpus. We consider multiple text corpora as the input data, for which there can be two applications of GANs: (1) the creation of consistent cross-corpus word embeddings given different word embeddings per corpus; (2) the generation of robust bag-of-words document embeddings for each corpora. We demonstrate our GAN models on real-world text data sets from different corpora, and show that embeddings from both models lead to improvements in supervised learning problems.

## 1 INTRODUCTION

Generative adversarial nets (GAN) (Goodfellow et al., 2014) belong to a class of generative models which are trainable and can generate artificial data examples similar to the existing ones. In a GAN model, there are two sub-models simultaneously trained: a generative model $\mathcal{G}$ from which artificial data examples can be sampled, and a discriminative model $\mathcal{D}$ which classifies real data examples and artificial ones from $\mathcal{G}$. By training $\mathcal{G}$ to maximize its generation power, and training $\mathcal{D}$ to minimize the generation power of $\mathcal{G}$, so that ideally there will be no difference between the true and artificial examples, a minimax problem can be established. The GAN model has been shown to closely replicate a number of image data sets, such as MNIST, Toronto Face Database (TFD), CIFAR-10, SVHN, and ImageNet (Goodfellow et al., 2014; Salimans et al., 2016).

The GAN model has been extended to text data in a number of ways. For instance, Zhang et al. (2016) applied a long-short term memory (LSTM) (Hochreiter & Schmidhuber, 1997) generator and approximated discretization to generate text data. Moreover, Li et al. (2017) applied the GAN model to generate dialogues, i.e. pairs of questions and answers. Meanwhile, the GAN model can also be applied to generate bag-of-words embeddings of text data, which focus more on key terms in a text document rather than the original document itself. Glover (2016) provided such a model with the energy-based GAN (Zhao et al., 2017).

To the best of our knowledge, there has been no literature on applying the GAN model to multiple corpora of text data. Multi-class GANs have been proposed (Liu & Tuzel, 2016; Mirza & Osindero, 2014), but a class in multi-class classification is not the same as multiple corpora. Because knowing the underlying corpus membership of each text document can provide better information on how the text documents are organized, and documents from the same corpus are expected to share similar topics or key words, considering the membership information can benefit the training of a text model from a supervised perspective. We consider two problems associated with training multi-corpus text data: (1) Given a separate set of word Mikolov et al. (2013b), how to obtain a better set of cross-corpus word embeddings from them? (2) How to incorporate the generation of document embeddings from different corpora in a single GAN model?

For the first problem, we train a GAN model which discriminates documents represented by different word embeddings, and train the cross-corpus word embedding so that it is similar to each existing word embedding per corpus. For the second problem, we train a GAN model which considers both cross-corpus and per-corpus "topics" in the generator, and applies a discriminator which considers each original and artificial document corpus. We also show that with sufficient training, the distribution of

the artificial document embeddings is equivalent to the original ones. Our work has the following contributions: (1) we extend GANs to multiple corpora of text data, (2) we provide applications of GANs to finetune word embeddings and to create robust document embeddings, and (3) we establish theoretical convergence results of the multi-class GAN model.

Section 2 reviews existing GAN models related to this paper. Section 3 describes the GAN models on training cross-corpus word embeddings and generating document embeddings for each corpora, and explains the associated algorithms. Section 4 presents the results of the two models on text data sets, and transfers them to supervised learning. Section 5 summarizes the results and concludes the paper.

## 2 Literature Review

In a GAN model, we assume that the data examples $\mathbf{x}$ are drawn from a distribution $p_{\mathbf{x}}(\cdot)$, and the artificial data examples $\mathcal{G}(\mathbf{z}) := \mathcal{G}(\mathbf{z}, \theta_g)$ are transformed from the noise distribution $\mathbf{z} \sim p_{\mathbf{z}}(\cdot)$. The binary classifier $\mathcal{D}(\cdot)$ outputs the probability of a data example (or an artificial one) being an original one. Because the probabilistic structure of a GAN can be unstable to train, the Wasserstein GAN (Arjovsky et al., 2017) is proposed which applies a 1-Lipschitz function as a discriminator.

We note that in many circumstances, data sets are obtained with supervised labels or categories, which can add explanatory power to unsupervised models such as the GAN. For instance, the CoGAN (Liu & Tuzel, 2016) considers pairs of data examples from different categories, and the weights of the first few layers (i.e. close to $\mathbf{z}$) are tied. Mirza & Osindero (2014) proposed the conditional GAN where the generator $\mathcal{G}$ and the discriminator $\mathcal{D}$ depend on the class label $y$. Salimans et al. (2016) applied the class labels for semi-supervised learning with an additional artificial class. However, all these models consider only images and do not produce word or document embeddings, therefore being different from our models.

For generating real text, Zhang et al. (2016) proposed textGAN in which the generator has an LSTM form, and a uni-dimensional convolutional neural network (Collobert et al., 2011; Kim, 2014) is applied as the discriminator. Also, a weighted softmax function is applied to make the argmax function differentiable. The focus of our work is to summarize information from longer documents, so we apply document embeddings such as the tf-idf to represent the documents rather than to generate real text.

For generating bag-of-words embeddings of text, Glover (2016) proposed a GAN model with the mean squared error of a de-noising autoencoder as the discriminator, and the output $\mathbf{x}$ is the one-hot word embedding of a document. Our models are different from this model because we consider tf-idf document embeddings for multiple text corpora in the deGAN model (Section 3.2), and weGAN (Section 3.1) can be applied to produce word embeddings. Also, we focus on robustness based on several corpora, while Glover (2016) assumed a single corpus.

For extracting word embeddings given text data, Mikolov et al. (2013a) proposed the word2vec model, for which there are two variations: the continuous bag-of-words (cBoW) model (Mikolov et al., 2013a), where the neighboring words are used to predict the appearance of each word; the skip-gram model, where each neighboring word is used individually for prediction. In GloVe (Pennington et al., 2014), a bilinear regression model is trained on the log of the word co-occurrence matrix. In these models, the weights associated with each word are used as the embedding. For obtaining document embeddings, the para2vec model (Le & Mikolov, 2014) adds per-paragraph vectors to train word2vec-type models, so that the vectors can be used as embeddings for each paragraph. A simpler approach by taking the average of the embeddings of each word in a document and output the document embedding is exhibited in Socher et al. (2013).

## 3 Models and Algorithms

Suppose we have a number of different corpora $\mathcal{C}^1, \ldots, \mathcal{C}^M$, which for example can be based on different categories or sentiments of text documents. We suppose that $\mathcal{C}^m = \{d_1^m, \ldots, d_{n_m}^m\}$, $m = 1, \ldots, M$, where each $d_i^m$ represents a document. The words in all corpora are collected in a

dictionary, and indexed from 1 to $V$. We name the GAN model to train cross-corpus word embeddings as "weGAN," where "we" stands for "word embeddings," and the GAN model to generate document embeddings for multiple corpora as "deGAN," where "de" stands for "document embeddings."

## 3.1 weGAN: Training cross-corpus word embeddings

We assume that for each corpora $\mathcal{C}^m$, we are given word embeddings for each word $v_1^m, \ldots, v_V^m \in \mathbb{R}^d$, where $d$ is the dimension of each word embedding. We are also given a classification task on documents that is represented by a parametric model $\mathcal{C}$ taking document embeddings as feature vectors. We construct a GAN model which combines different sets of word embeddings $\mathcal{V}^m := \{v_i^m\}_{i=1}^V$, $m = 1, \ldots, M$, into a single set of word embeddings $\mathcal{G} := \{v_i^0\}_{i=1}^V$. Note that $\mathcal{V}^1, \ldots, \mathcal{V}^M$ are given but $\mathcal{G}$ is trained. Here we consider $\mathcal{G}$ as the generator, and the goal of the discriminator is to distinguish documents represented by the original embeddings $\mathcal{V}^1, \ldots, \mathcal{V}^M$ and the same documents represented by the new embeddings $\mathcal{G}$.

Next we describe how the documents are represented by a set of embeddings $\mathcal{V}^1, \ldots, \mathcal{V}^M$ and $\mathcal{G}$. For each document $d_i^m$, we define its document embedding with $\mathcal{V}^m$ as $g_i^m := f(d_i^m, \mathcal{V}^m)$, where $f(\cdot)$ can be any mapping. Similarly, we define the document embedding of $d_i^m$ with $\mathcal{G}$ as follows, with $\mathcal{G} = \{v_j^0\}_{j=1}^V$ trainable $f_{\mathcal{G}}(d_i^m) := f(d_i^m, \mathcal{G})$. In a typical example, word embeddings would be based on word2vec or GLoVe. Function $f$ can be based on tf-idf, i.e. $f(d_i^m, \mathcal{V}) = \sum_{j=1}^V t_{ij}^m v_j^m$ where $v_j^m$ is the word embedding of the $j$-th word in the $m$-th corpus $\mathcal{C}^m$ and $t_i^m = (t_{i1}^m, \ldots, t_{iV}^m)$ is the tf-idf representation of the $i$-th document $d_i^m$ in the $m$-th corpus $\mathcal{C}^m$.

To train the GAN model, we consider the following minimax problem

$$\min_{\mathcal{C}, \mathcal{G}} \max_{\mathcal{D}} \left\{ \sum_{m=1}^M \sum_{i=1}^{n_m} [\log(\mathcal{D}(g_i^m)) + \log(1 - \mathcal{D}(f_{\mathcal{G}}(d_i^m))) - \log(e_{k_i^m}^T \mathcal{C}(f_{\mathcal{G}}(d_i^m)))] \right\}, \quad (1)$$

where $\mathcal{D}$ is a discriminator of whether a document is original or artificial. Here $k_i^m$ is the label of document $d_i^m$ with respect to classifier $\mathcal{C}$, and $e_{k_i^m}$ is a unit vector with only the $k_i^m$-th component being one and all other components being zeros. Note that $\log(e_{k_i^m}^T \mathcal{C}(f_{\mathcal{G}}(d_i^m)))$ is equivalent to $KL(e_{k_i^m} \| \mathcal{C}(f_{\mathcal{G}}(d_i^m)))$, but we use the former notation due to its brevity.

The intuition of problem (3) is explained as follows. First we consider a discriminator $\mathcal{D}$ which is a feedforward neural network (FFNN) with binary outcomes, and classifies the document embeddings $\{f_{\mathcal{G}}(d_i^m)\}_{i=1}^{n_m}{}_{m=1}^M$ against the original document embeddings $\{g_i^m\}_{i=1}^{n_m}{}_{m=1}^M$. Discriminator $\mathcal{D}$ minimizes this classification error, i.e. it maximizes the log-likelihood of $\{f_{\mathcal{G}}(d_i^m)\}_{i=1}^{n_m}{}_{m=1}^M$ having label 0 and $\{g_i^m\}_{i=1}^{n_m}{}_{m=1}^M$ having label 1. This corresponds to

$$\sum_{m=1}^M \sum_{i=1}^{n_m} [\log(\mathcal{D}(g_i^m)) + \log(1 - \mathcal{D}(f_{\mathcal{G}}(d_i^m)))]. \quad (2)$$

For the generator $\mathcal{G}$, we wish to minimize (3) against $\mathcal{G}$ so that we can apply the minimax strategy, and the combined word embeddings $\mathcal{G}$ would resemble each set of word embeddings $\mathcal{V}^1, \ldots, \mathcal{V}^M$. Meanwhile, we also consider classifier $\mathcal{C}$ with $K$ outcomes, and associates $d_i^m$ with label $k_i^m$, so that the generator $\mathcal{G}$ can learn from the document labeling in a semi-supervised way.

If the classifier $\mathcal{C}$ outputs a $K$-dimensional softmax probability vector, we minimize the following against $\mathcal{G}$, which corresponds to (3) given $\mathcal{D}$ and $\mathcal{C}$:

$$\sum_{m=1}^M \sum_{i=1}^{n_m} [\log(1 - \mathcal{D}(f_{\mathcal{G}}(d_i^m))) - \log(e_{k_i^m}^T \mathcal{C}(f_{\mathcal{G}}(d_i^m)))]. \quad (3)$$

For the classifier $\mathcal{C}$, we also minimize its negative log-likelihood

$$- \sum_{m=1}^M \sum_{i=1}^{n_m} \log(e_{k_i^m}^T \mathcal{C}(f_{\mathcal{G}}(d_i^m))). \quad (4)$$

Assembling (4-6) together, we retrieve the original minimax problem (3).

We train the discriminator and the classifier, $\{\mathcal{D}, \mathcal{C}\}$, and the combined embeddings $\mathcal{G}$ according to (4-6) iteratively for a fixed number of epochs with the stochastic gradient descent algorithm, until the discrimination and classification errors become stable.

Figure 1 illustrates the weGAN model. The algorithm for weGAN is summarized in Algorithm 1 in the appendix.

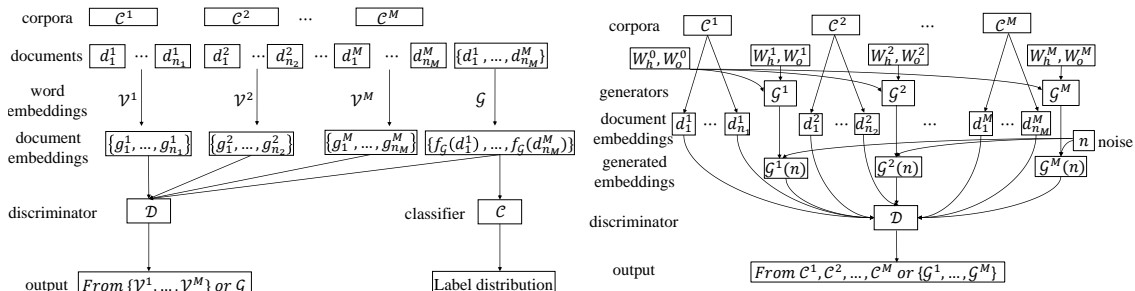

Figure 1: Model structure of weGAN.   Figure 2: Model structure of deGAN.

## 3.2 deGAN: Generating document embeddings for multi-corpus text data

In this section, our goal is to generate document embeddings which would resemble real document embeddings in each corpus $\mathcal{C}^m$, $m = 1, \ldots, M$. We construct $M$ generators, $\mathcal{G}^1, \ldots, \mathcal{G}^M$ so that $\mathcal{G}^m$ generate artificial examples in corpus $\mathcal{C}^m$. As in Section 3.1, there is a certain document embedding such as tf-idf, bag-of-words, or para2vec. Let $\mathcal{G} = \{\mathcal{G}_1, \ldots, \mathcal{G}_M\}$. We initialize a noise vector $n = (n_1, \ldots, n_{d_n}) \in \mathbb{R}^{d_n}$, where $n_1, \ldots, n_{d_n} \overset{iid}{\sim} \mathcal{N}$, and $\mathcal{N}$ is any noise distribution.

For a generator $\mathcal{G}^m = \{W_h^m, W_h^0, W_o^m, W_o^0\}$ represented by its parameters, we first map the noise vector $n$ to the hidden layer, which represents different topics. We consider two hidden vectors, $h^0$ for general topics and $h^m$ for specific topics per corpus, $h^m = a_1(W_h^m n)$, $h^0 = a_1(W_h^0 n)$. Here $a_1(\cdot)$ represents a nonlinear activation function. In this model, the bias term can be ignored in order to prevent the "mode collapse" problem of the generator. Having the hidden vectors, we then map them to the generated document embedding with another activation function $a_2(\cdot)$,

$$o_m = a_2(W_o^m h^m + w_o^0 h^0). \tag{5}$$

To summarize, we may represent the process from noise to the document embedding as $G^m(n) = a_2(W_o^m a_1(W_h^m n) + w_o^0 a_1(W_h^0 n))$. Given the generated document embeddings $\mathcal{G}^1(n), \ldots, \mathcal{G}^M(n)$, we consider the following minimax problem to train the generator $\mathcal{G}$ and the discriminator $\mathcal{D}$:

$$\min_{\mathcal{G}} \sum_{m=1}^{M} E_n \log \left\{ e_{M+m}^T \mathcal{D}_{\mathcal{G}}^*(\mathcal{G}^m(n)) / \ [e_{M+m}^T \mathcal{D}_{\mathcal{G}}^*(\mathcal{G}^m(n)) + e_m^T \mathcal{D}_{\mathcal{G}}^*(\mathcal{G}^m(n))] \right\}, \tag{6}$$

where

$$\mathcal{D}_{\mathcal{G}}^* \in \arg\max_{\mathcal{D}} \sum_{m=1}^{M} E_{d^m \sim p_m} \left[ \log(e_m^T \mathcal{D}(d^m)) \right] + \sum_{m=1}^{M} E_n [\log(e_{M+m}^T \mathcal{D}(\mathcal{G}^m(n)))]. \tag{7}$$

Here we assume that any document embedding $d^m$ in corpus $\mathcal{C}^m$ is a sample with respect to the probability density $p_m$. Note that when $M = 1$, the discriminator part of our model is equivalent to the original GAN model.

To explain (10), first we consider the discriminator $\mathcal{D}$. Because there are multiple corpora of text documents, here we consider $2M$ categories as output of $\mathcal{D}$, from which categories $1, \ldots, M$ represent the original corpora $\mathcal{C}^1, \ldots, \mathcal{C}^M$, and categories $M + 1, \ldots, 2M$ represent the generated document embeddings (e.g. bag-of-words) from $\mathcal{G}^1, \ldots, \mathcal{G}^M$. Assume the discriminator $\mathcal{D}$, a feedforward neural network, outputs the distribution of a text document being in each category. We maximize the

log-likelihood of each document being in the correct category against $\mathcal{D}$

$$\sum_{m=1}^{M} E_{p_m} \left[\log(e_m^T \mathcal{D}(d^m))\right] + \sum_{m=1}^{M} E_n[\log(e_{M+m}^T \mathcal{D}(\mathcal{G}^m(n)))]. \tag{8}$$

Such a classifier does not only classifies text documents into different categories, but also considers $M$ "fake" categories from the generators. When training the generators $\mathcal{G}^1, \ldots, \mathcal{G}^M$, we minimize the following which makes a comparison between the $m$-th and $(M+m)$-th categories

$$\sum_{m=1}^{M} E_n \log \frac{e_{M+m}^T \mathcal{D}(\mathcal{G}^m(n))}{e_{M+m}^T \mathcal{D}(\mathcal{G}^m(n)) + e_m^T \mathcal{D}(\mathcal{G}^m(n))}. \tag{9}$$

The intuition of (13) is that for each generated document embedding $\mathcal{G}^m(n)$, we need to decrease $e_{M+m}^T \mathcal{D}(\mathcal{G}^m(n))$, which is the probability of the generated embedding being correctly classified, and increase $e_m^T \mathcal{D}(\mathcal{G}^m(n))$, which is the probability of the generated embedding being classified into the target corpus $\mathcal{C}^m$. The ratio in (13) reflects these two properties.

We iteratively train (12) and (13) until the classification error of $\mathcal{D}$ becomes stable. Figure 2 illustrates the deGAN model. The algorithm for deGAN is summarized in Algorithm 2 in the appendix.

We next show that from (10), the distributions of the document embeddings from the optimal $\mathcal{G}^1, \ldots, \mathcal{G}^M$ are equal to the data distributions of $\mathcal{C}^1, \ldots, \mathcal{C}^M$, which is a generalization of Goodfellow et al. (2014) to the multi-corpus scenario. The proof of Proposition 1 is in the appendix.

**Proposition 1**. Let us assume that the random variables $d^1, \ldots, d^M$ are continuous with probability density $p_1, \ldots, p_M$ which have bounded support $\mathcal{X}$; $n$ is a continuous random variable with bounded support and activations $a_1$ and $a_2$ are continuous; and that $\mathcal{G}^{1*}, \ldots, \mathcal{G}^{M*}$ are solutions to (10). Then $q_1^*, \ldots, q_M^*$, the probability density of the document embeddings from $\mathcal{G}^{1*}, \ldots, \mathcal{G}^{M*}$, are equal to $p_1, \ldots, p_M$. □

# 4 Experiments

In the experiments, we consider four data sets, two of them newly created and the remaining two already public: CNN, TIME, 20 Newsgroups (in the appendix), and Reuters-21578 (in the appendix). The code and the two new data sets are available at github.com/deeplearning2018/emgan. For the pre-processing of all the documents, we transformed all characters to lower case, stemmed the documents, and ran the word2vec model on each corpora to obtain word embeddings with a size of 300. In all subsequent models, we only consider the most frequent 5,000 words across all corpora in a data set.

The document embedding in weGAN is the tf-idf weighted word embedding transformed by the $\tanh$ activation, i.e. $f(d_i^m, \mathcal{V}^m) = \tanh\left(\sum_{j=1}^{V} t_{ij}^m v_j^m\right)$. For deGAN, we use $L^1$-normalized tf-idf as the document embedding because it is easier to interpret than the transformed embedding in (20).

For weGAN, the cross-corpus word embeddings are initialized with the word2vec model trained from all documents. For training our models, we apply a learning rate which increases linearly from 0.01 to 1.0 and train the models for 100 epochs with a batch size of 50 per corpus. The classifier $\mathcal{C}$ has a single hidden layer with 50 hidden nodes, and the discriminator with a single hidden layer $\mathcal{D}$ has 10 hidden nodes. All these parameters have been optimized. For the labels $k_i^m$ in (8), we apply corpus membership of each document.

For the noise distribution $\mathcal{N}$ for deGAN, we apply the uniform distribution $U(-1, 1)$. In (14) for deGAN, $a_1 = \tanh$ and $a_2 = softmax$ so that the model outputs document embedding vectors which are comparable to $L^1$-normalized tf-idf vectors for each document. For the discriminator $\mathcal{D}$ of deGAN, we apply the word2vec embeddings based on all corpora to initialize its first layer, followed by another hidden layer of 50 nodes. For the discriminator $\mathcal{D}$, we apply a learning rate of 0.1, and for the generator $\mathcal{G}$, we apply a learning rate of 0.001, because the initial training phase of deGAN can be unstable. We also apply a batch size of 50 per corpus. For the softmax layers of deGAN, we initialize them with the log of the topic-word matrix in latent Dirichlet allocation (LDA) (Blei et al., 2003) in order to provide intuitive estimates.

| | w2v-RI | weGAN-RI |
|---|---|---|
| mean | 67.88% | **68.45%** |
| sd. | 0.02% | 0.01% |
| | w2v-accuracy | weGAN-accuracy |
| mean | 92.05% | **92.36%** |
| sd. | 0.06% | 0.03% |

Table 1: A comparison between word2vec and weGAN in terms of the Rand index and the classification accuracy for the CNN data set.

| | | |
|---|---|---|
| Obama | w2v | Bush Trump Kerry Abe Netanyahu Rouhani Erdogan he Karzai Tillerson |
| Obama | weGAN | Trump Bush Kerry Abe Netanyahu Erdogan Tillerson he Carter Rouhani |
| Trump | w2v | Obama Pence Erdogan Bush Duterte he Sanders Macron Christie Tillerson |
| Trump | weGAN | Obama Pence Bush Christie Sanders Clinton Erdogan Tillerson Macron Duterte |
| U.S. | w2v | US Pentagon United Iranian NATO Turkish Qatar Iran British UAE |
| U.S. | weGAN | US Pentagon United Iranian NATO Turkish Iran Qatar American UAE |

Table 2: Synonyms of "Obama," "Trump," and "U.S." before and after weGAN training for the CNN data set.

For weGAN, we consider two metrics for comparing the embeddings trained from weGAN and those trained from all documents: (1) applying the document embeddings to cluster the documents into $M$ clusters with the K-means algorithm, and calculating the Rand index (RI) (Rand, 1971) against the original corpus membership; (2) finetuning the classifier $\mathcal{C}$ and comparing the classification error against an FFNN of the same structure initialized with word2vec (w2v). For deGAN, we compare the performance of finetuning the discriminator of deGAN for document classification, and the performance of the same FFNN. Each supervised model is trained for 500 epochs and the validation data set is used to choose the best epoch.

## 4.1 The CNN data set

In the CNN data set, we collected all news links on www.cnn.com in the GDELT 1.0 Event Database from April 1st, 2013 to July 7, 2017. We then collected the news articles from the links, and kept those belonging to the three largest categories: "politics," "world," and "US." We then divided these documents into 21,674 training documents, from which 2,708 validation documents are held out, and 5,420 testing documents.

We hypothesize that because weGAN takes into account document labels in a semi-supervised way, the embeddings trained from weGAN can better incorporate the labeling information and therefore, produce document embeddings which are better separated. The results are shown in Table 1 and averaged over 5 randomized runs. Performing the Welch's t-test, both changes after weGAN training are statistically significant at a 0.05 significance level. Because the Rand index captures matching accuracy, we observe from Table 1 that weGAN tends to improve both metrics.

Meanwhile, we also wish to observe the spatial structure of the trained embeddings, which can be explored by the synonyms of each word measured by the cosine similarity. On average, the top 10 synonyms of each word differ by 0.22 word after weGAN training, and 20.7% of all words have different top 10 synonyms after training. Therefore, weGAN tends to provide small adjustments rather than structural changes. Table 2 lists the 10 most similar terms of three terms, "Obama," "Trump," and "U.S.," before and after weGAN training, ordered by cosine similarity.

We observe from Table 2 that for "Obama," "Trump" and "Tillerson" are more similar after weGAN training, which means that the structure of the weGAN embeddings can be more up-to-date. For "Trump," we observe that "Clinton" is not among the synonyms before, but is after, which shows that the synonyms after are more relevant. For "U.S.," we observe that after training, "American" replaces "British" in the list of synonyms, which is also more relevant.

We next discuss deGAN. In Table 3, we compare the performance of finetuning the discriminator of deGAN for document classification, and the performance of the FFNN initialized with word2vec. The change is also statistically significant at the 0.05 level. From Table 3, we observe that deGAN improves the accuracy of supervised learning.

|        | w2v-accuracy | deGAN-accuracy |
|--------|:------------:|:--------------:|
| mean   | 92.05%       | **92.29%**     |
| sd.    | 0.06%        | 0.09%          |

Table 3: A comparison between word2vec and deGAN in terms of the accuracy for the CNN data set.

| politics | original | India US Carter defense Indian relationship China said military two |
|----------|----------|---------------------------------------------------------------------|
| politics | deGAN    | year US meeting used read security along building worth foreign |
| world    | original | Turkey Turkish Attack ISIS said Kurdish Erdogan group bomb report |
| world    | deGAN    | cut company get lot made code could Steve items may road block phone |
| US       | original | climate change year study according says country temperatures average |
| US       | deGAN    | area efforts volunteers town weapons shot local nearly department also |

Table 4: Bag-of-words representations of original and artificial text in the CNN data set.

To compare the generated samples from deGAN with the original bag-of-words, we randomly select one record in each original and artificial corpus. The records are represented by the most frequent words sorted by frequency in descending order where the stop words are removed. The bag-of-words embeddings are shown in Table 4.

From Table 4, we observe that the bag-of-words embeddings of the original documents tend to contain more name entities, while those of the artificial deGAN documents tend to be more general. There are many additional examples not shown here with observed artificial bag-of-words embeddings having many name entities such as "Turkey," "ISIS," etc. from generated documents, e.g. "Syria eventually ...list...'

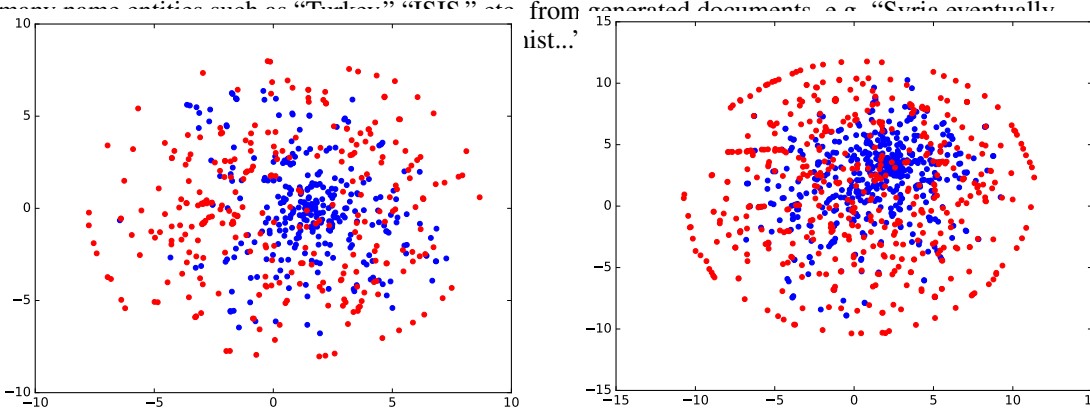

Figure 3: 2-d representation of original (red) and artificial (blue) examples in the CNN data set.

Figure 4: 2-d representation of original (red) and artificial (blue) examples in the TIME data set.

We also perform dimensional reduction using t-SNE (van der Maaten & Hinton, 2008), and plot 100 random samples from each original or artificial category. The original samples are shown in red and the generated ones are shown in blue in Figure 3. We do not further distinguish the categories because there is no clear distinction between the three original corpora, "politics," "world," and "US."

We observe that the original and artificial examples are generally mixed together and not well separable, which means that the artificial examples are similar to the original ones. However, we also observe that the artificial samples tend to be more centered and have no outliers (represented by the outermost red oval).

## 4.2   The TIME data set

In the TIME data set, we collected all news links on time.com in the GDELT 1.0 Event Database from April 1st, 2013 to July 7, 2017. We then collected the news articles from the links, and kept

those belonging to the five largest categories: "Entertainment," "Ideas," "Politics," "US," and "World." We divided these documents into 12,286 training documents, from which 1,535 validation documents are held out, and 3,075 testing documents.

Table 5 compares the clustering results of word2vec and weGAN, and the classification accuracy of an FFNN initialized with word2vec, finetuned weGAN, and finetuned deGAN. The results in Table 5 are the counterparts of Table 1 and Table 3 for the TIME data set. The differences are also significant at the 0.05 level.

|  | w2v-RI | weGAN-RI |
|---|---|---|
| mean | 70.96% | **71.14%** |
| sd. | 0.02% | 0.02% |
| w2v-accur. | weGAN-accur. | deGAN-accur. |
| 83.79% | 84.76% | **85.38%** |
| 0.17% | 0.08% | 0.11% |

Table 5: A comparison between word2vec, weGAN, and deGAN in terms of the Rand index and the classification accuracy for the TIME data set.

| Obama | w2v | Trump Bush Xi Erdogan Rouhani Reagan Hollande Duterte Abe Jokowi |
|---|---|---|
| Obama | weGAN | Trump Bush Xi Erdogan Reagan Rouhani Hollande Abe Jokowi Duterte |
| Trump | w2v | Obama Erdogan Rubio Duterte Bush Putin Sanders Xi Macron Pence |
| Trump | weGAN | Obama Erdogan Rubio Bush Sanders Putin Duterte Xi Macron Pence |
| U.S. | w2v | NATO Iran Japan Pentagon Russia Pakistan Tehran EU Ukrainian Moscow |
| U.S. | weGAN | NATO Pentagon Iran Japan Russia Tehran Pakistan EU Ukrainian Moscow |

Table 6: Synonyms of "Obama," "Trump," and "U.S." before and after weGAN training for the TIME data set.

From Table 5, we observe that both GAN models yield improved performance of supervised learning. For weGAN, on an average, the top 10 synonyms of each word differ by 0.27 word after weGAN training, and 24.8% of all words have different top 10 synonyms after training. We also compare the synonyms of the same common words, "Obama," "Trump," and "U.S.," which are listed in Table 6. In the TIME data set, for "Obama," "Reagan" is ranked slightly higher as an American president. For "Trump," "Bush" and "Sanders" are ranked higher as American presidents or candidates. For "U.S.," we note that "Pentagon" is ranked higher after weGAN training, which we think is also reasonable because the term is closely related to the U.S. government.

For deGAN, we also compare the original and artificial samples in terms of the highest probability words, which is shown in Table 7. We also perform dimensional reduction using t-SNE for 100 examples per corpus and plot them in Figure 4. All these figures and tables show results similar to Section 4.1.

| Entertainment | original | show London attack people proud open going right according way home |
|---|---|---|
| Entertainment | deGAN | music actor Michael John song going meeting James produced pop vocal |
| Ideas | original | American would service national young country year serve security world |
| Ideas | deGAN | city project part development grand bear often west new status high agents |
| Politics | original | Assange embassy BBC Swedish charges told authorities officials Sweden |

| Politics | deGAN | members present national committee party Paul sign Trump removed brief |
|---|---|---|
| US | original | Charleston many Carolina South funeral hand wrote political words |
| US | deGAN | Davis head board man relationship recent Sunday stone fire wrote gay well |
| world | original | Erdogan Turkey political power government two leaders minister AKP |
| world | deGAN | suffering like know old violence local daily young interest three first man |

Table 7: Bag-of-words representations of original and artificial text in the TIME data set.

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

# A    Algorithms for weGAN and deGAN

**Algorithm 1 (for weGAN).**

1. Train $\mathcal{G}$ based on $f$ from all corpora $\mathcal{C}^1, \ldots, \mathcal{C}^M$.
2. Randomly initialize the weights and biases of the classifier $\mathcal{C}$ and discriminator $\mathcal{D}$.
3. Until maximum number of iterations reached
   (a) Update $\mathcal{C}$ and $\mathcal{D}$ according to (4) and (6) given a mini-batch $S_1$ of training examples $\{d_i^m\}_{i,m}$.
   (b) Update $\mathcal{G}$ according to (5) given a mini-batch $S_2$ of training examples $\{d_i^m\}_{i,m}$.
4. Output $\mathcal{G}$ as the cross-corpus word embeddings.

**Algorithm 2 (for deGAN).**

1. Randomly initialize the weights of $\mathcal{G}^1, \ldots, \mathcal{G}^M$.
2. Initialize the discriminator $\mathcal{D}$ with the weights of the first layer (which takes document embeddings as the input) initialized by word embeddings, and other parameters randomly initialized.
3. Until maximum number of iterations reached
   (a) Update $\mathcal{D}$ according to (12) given a mini-batch of training examples $d_i^m$ and samples from noise $n$.
   (b) Update $\mathcal{G}^1, \ldots, \mathcal{G}^M$ according to (13) given a mini-batch of training examples $d_i^m$ and samples form noise $n$.
4. Output $\mathcal{G}^1, \ldots, \mathcal{G}^M$ as generators of document embeddings and $\mathcal{D}$ as a corpus classifier.

# B    Proof of Proposition 1

Since $\mathcal{X}$ is bounded, all of the integrals exhibited next are well-defined and finite. Since $n$, $a_1$, and $a_2$ are continuous, it follows that for any parameters, $\mathcal{G}^m(n)$ is a continuous random variable with probability density $q_m$ with finite support. From (11),

$$
\begin{aligned}
\mathcal{D}_{\mathcal{G}}^* = \arg\max_{\mathcal{D}} \Bigg\{ &\sum_{m=1}^{M} \int p_m(x) \log(e_m^T \mathcal{D}(x)) dx \\
&+ \sum_{m=1}^{M} \int q_m(x) \log(e_{M+m}^T \mathcal{D}(x)) dx \Bigg\} \\
= \arg\max_{\mathcal{D}} \Bigg\{ &\int \sum_{m=1}^{M} p_m(x) \log(e_m^T \mathcal{D}(x)) \\
&+ \sum_{m=1}^{M} q_m(x) \log(e_{M+m}^T \mathcal{D}(x)) dx \Bigg\}.
\end{aligned}
\tag{10}
$$

This problem reduces to

$$
\max_{b_1, \ldots, b_m} \sum_{m=1}^{M} a_m \log b_m \text{ subject to } \sum_{m=1}^{M} b_m = 1,
\tag{11}
$$

the solution of which is $b_m^* = a_m / \sum_{m=1}^{M} a_m$, $m = 1, \ldots, M$. Therefore, the solution to (15) is

$$
\mathcal{D}_{\mathcal{G}}^*(x) = \frac{(p_1(x), \ldots, p_M(x), q_1(x), \ldots, q_M(x))}{\sum_{m=1}^{M} p_m(x) + \sum_{m=1}^{M} q_m(x)}.
\tag{12}
$$

We then obtain from (10) that

$$
\begin{aligned}
q_1^*, \ldots, q_M^* \in \arg\min_{q_1,\ldots,q_M} \sum_{m=1}^{M} \int q_m(x) \cdot \log\left[\frac{q_m(x)}{q_m(x) + p_m(x)}\right] dx \\
= \arg\min_{q_1,\ldots,q_M} -M\log 2 + \sum_{m=1}^{M} \int q_m(x)\log\left[\frac{q_m(x)}{(q_m(x) + p_m(x))/2}\right] dx \\
= \arg\min_{q_1,\ldots,q_M} -M\log 2 + \sum_{m=1}^{M} KL(q_m\|(q_m + p_m)/2).
\end{aligned}
$$

From non-negativity of the Kullback-Leibler divergence, we conclude that

$$
(q_1^*, \ldots, q_M^*) = (p_1, \ldots, p_M). \qquad \square
$$

## C  The 20 Newsgroups data set

The 20 Newsgroups data set is a collection of news documents with 20 categories. To reduce the number of categories so that the GAN models are more compact and have more samples per corpus, we grouped the documents into 6 super-categories: "religion," "computer," "cars," "sport," "science," and "politics" ("misc" is ignored because of its noisiness). We considered each super-category as a different corpora. We then divided these documents into $10{,}708$ training documents, from which $1{,}335$ validation documents are held out, and $7{,}134$ testing documents. We train weGAN and deGAN in the the beginning of Section 4, except that we use a learning rate of $0.01$ for the discriminator in deGAN to stabilize the cost function. Table 8 compares the clustering results of word2vec and weGAN, and the classification accuracy of the FFNN initialized with word2vec, finetuned weGAN, and finetuned deGAN. All comparisons are statistically significant at the $0.05$ level. The other results are similar to the previous two data sets and are thereby omitted here.

| | w2v-RI | weGAN-RI |
|---|---|---|
| mean | 76.14% | **76.74%** |
| sd. | 0.02% | 0.08% |
| w2v-accur. | weGAN-accur. | deGAN-accur. |
| 87.34% | **89.90%** | 89.32% |
| 0.04% | 0.02% | 0.15% |

Table 8: A comparison between word2vec, weGAN, and deGAN in terms of the Rand index and the classification accuracy for the 20 Newsgroups data set.

## D  The Reuters-21578 data set

The Reuters-21578 data set is a collection of newswire articles. Because the data set is highly skewed, we considered the eight categories with more than 100 training documents: "earn," "acq," "crude," "trade," "money-fx," "interest," "money-supply," and "ship." We then divided these documents into $5{,}497$ training documents, from which $692$ validation documents are held out, and $2{,}207$ testing documents. We train weGAN and deGAN in the same way as in the 20 Newsgroups data set. Table 9 compares the clustering results of word2vec and weGAN, and the classification accuracy of the FFNN initialized with word2vec, finetuned weGAN, and finetuned deGAN. All comparisons are statistically significant at the $0.05$ level except the Rand index. The other results are similar to the CNN and TIME data sets and are thereby omitted here.

|  | w2v-RI | weGAN-RI |
|---|---|---|
| mean | 71.28% | **71.43%** |
| sd. | 0.26% | 0.07% |

| w2v-accur. | weGAN-accur. | deGAN-accur. |
|---|---|---|
| 92.86% | **95.10%** | 94.86% |
| 0.09% | 0.10% | 0.10% |

Table 9: A comparison between word2vec, weGAN, and deGAN in terms of the Rand index and the classification accuracy for the Reuters-21578 data set.

