# OpenReview forum: "Generative Adversarial Nets for Multiple Text Corpora"
_ICLR.cc/2020/Conference — Reject_

### Official Review · AnonReviewer1 · 2019-10-22
**Official Blind Review #1**

**Rating:** 3

**Review:**

This work proposes two models: (1) A model to learn word embeddings from different corpus. (2) A model to generate robust bag-of-words document embeddings. This topic may not be novel enough, considering the current development in GAN and domain adaptation. Some questions:
-	For the word embedding model, why it is called GAN? It seems that nothing is generated.
-	Further, for the word embedding model, the notation is over-complicated. This model is very similar to the domain adversarial network (DANN). This is my understanding and please correct me if I am inaccurate: A classifier is trained to distinguish the domains/corpus, while the word encoder tries to learn shared features across domains/corpus. The sub- and super-scripts are confusing.
-	For the deGAN, only generating bag-of-words may not be that attracting. Further, the idea is also similar to many existing works in domain adaptation. Each domain/corpus has unique components, while all domains/corpus also share similar features. The final output is just a combination of the shared components and the unique features. This idea is straightforward. It would be much better if the author can generate natural language, instead of just bag-of-words.
-	Please fix the formatting issues. For instance, Figure 3 and Figure 4 cover the text. ‘To explain (10), first we consider…’. I think here should be (6).
-	More baseline competitors are preferred.


**Experience Assessment:**

I have read many papers in this area.

**Review Assessment: Checking Correctness Of Derivations And Theory:**

I carefully checked the derivations and theory.

**Review Assessment: Checking Correctness Of Experiments:**

I assessed the sensibility of the experiments.

**Review Assessment: Thoroughness In Paper Reading:**

I read the paper at least twice and used my best judgement in assessing the paper.

---

### Official Review · AnonReviewer3 · 2019-10-25
**Official Blind Review #3**

**Rating:** 3

**Review:**

Summary

The paper proposes extensions of Generative Adversarial Networks to modeling multiple text corpora. Concretely, the paper looks at two problems: 1) given independently pretrained word embeddings from K corpora, finding a common word embdding, 2) extracting document representations from a discriminator of a GAN trained to generate tf-idf vectors. Preliminary experiments show that the proposed approaches outperform baseline classifiers trained with word2vec.


Strengths
+ The paper clearly mentions all the experimental details
+ The paper has a nice set of qualitative examples that probe what the proposed model is learning


Weaknesses

* It is not clear what problem the paper is trying to solve. Is the goal to get better document representations, in which case why do we think that using a GAN is a good idea. Learning a generative model to use representations for a downstream task seems like a pretty roundabout way of doing things (if we dont care about generating anything in the first place). Further, if better document representations are desired then the paper should compare to works like Bert (Devlin et.al. [1]).

* Sec. 3.1: Calling the proposed weGAN model a GAN seems a bit inconsistent/ wrong. The proposed model is not a generative adversarial network, there is no sampling of a noise or a notion of a generative distribution. The real data, similar to the image case is a bunch of samples from the data distribution (where the stochasticity comes not from the word embeddings but from the tf-idf of thte document), but the generator also uses the tf-idf representation, so essentially both the generator as well as the real world data use the document as an input. Hence it feels like a somewhat odd model formulation, which would be nice to clarify / explain.

* Further, it is not clear why one would want to train “cross-corpus” embeddings in the way describeed in Sec. 3. 1. Why not just train word2vec on the union of all the corpora (instead of per-corpus) and use it as the word representation? What are the conditions in which one would not want to do the common word representation?

* Sec. 3.2: Why not formulate modeling multiple corpora in the spirit of CoGAN (Liu and Tuzel, 2016), by getting M discriminators (potentially with parameter sharing) to solve binary classification tasks as opposed to the 2M way classification problem? Atleast a comparison to an approach like this seems warranted. Further, it is not clear why one would want to generate tf-idf vectors. It seems like all the experiments are around representation learning and classification as opposed to generating or evaluating tf-idf vectors for documents.

* As a side note, for the experiments concerning deGAN vs the baseline, it would be nice to check what happens when the last classification layer is not initialized with LDA parameters. Since that is an additional source of information.

References
[1]: Devlin, Jacob, Ming-Wei Chang, Kenton Lee, and Kristina Toutanova. 2018. “BERT: Pre-Training of Deep Bidirectional Transformers for Language Understanding.” arXiv [cs.CL]. arXiv. http://arxiv.org/abs/1810.04805.


**Experience Assessment:**

I have published one or two papers in this area.

**Review Assessment: Checking Correctness Of Derivations And Theory:**

I did not assess the derivations or theory.

**Review Assessment: Checking Correctness Of Experiments:**

I assessed the sensibility of the experiments.

**Review Assessment: Thoroughness In Paper Reading:**

I read the paper at least twice and used my best judgement in assessing the paper.

---

### Official Review · AnonReviewer4 · 2019-10-28
**Official Blind Review #4**

**Rating:** 1

**Review:**

The paper proposes to use Generative Adversarial Networks (GANs) in the context of natural language processing and introduces two models for generating document embeddings. The first model, weGAN, aggregates multiple sets of single-corpus word2vec embeddings into one set of cross-corpus word representations; document embeddings are a function of these updated word embeddings. The second model, deGAN, side-steps word-level embeddings and directly generates document-level representations. For both models, the real examples come from word2vec and tf-idf, while the artificial examples are the ones generated by the network. The authors show that their document embeddings are better than the word2vec/tf-idf baseline at clustering documents according to the corpus they originate from.

While the context in which GANs are used is novel and creative, the reasons for rejection outweigh the positives. The main reason is the fact that the paper is outdated by two/three years. There is no mention of the shift towards contextual embeddings that has been happening in the last few years (InferSent, ELMo, BERT) and that became ubiquitous in NLP.  Instead, all comparisons are against word2vec (published in 2013) that is no longer a relevant baseline, even for non-contextual word embeddings. In fact, the newest cited papers are from 2017. The Github link in the paper was last updated on 23rd December 2017. My objection is not that this work was started a long time ago, but that it has not been updated in years. In the lack of proper comparison against more novel techniques, it is hard to estimate whether the reported gains over word2vec are meaningful.

Another argument for rejection is the fact that the experimental section does not make a convincing case that the positive results matter beyond the very specific datasets and tasks selected.

Before commenting in more detail on the selection of datasets, tasks and metrics, I want to document my understanding of the experimental setup; I found it non-trivial to understand what constitutes a "single corpus" in the experiments. My mental model is the following: the paper operates on four datasets (CNN, TIME, 20 Newsgroups, and Reuters-21578). These are four different experiments, and the four datasets never interact. The cross-corpus property of the model is exercised within each of these datasets, which have a natural grouping of topics; for instance, CNN is divided into "US", "world", and "politics". In other words, a cross-corpus CNN model is trained on three corpora: US, world, and politics. With this mental model of the experimental setup, the experiments raise a series of questions:

1) Do the selected datasets / classification tasks matter in the grad scheme of NLP?
The end-task is to classify the corpus of origin for each document (for instance, for the CNN experiment, the model has to decide between US, world, and politics). If I understand correctly, the weGAN model jointly trained its word embeddings with this objective. As a final step, the classifier was additionally fine-tuned on this task. So it is not very surprising that weGAN is better at clustering documents than word2vec, since it was explicitly trained to do so. It would have been more informative to see weGAN performing better on a task that was solely fine-tuned on (e.g. some sentiment classification task that was not part of the training objective when embeddings were trained). In that case, one could argue that the enhanced embeddings can be used as an out-of-the box tool for a lot of different tasks. Another option would be evaluating on the same document membership task, but on datasets that were not seen during training.

Additionally, the fact that the authors evaluated on *new* datasets with a *new* task distances them even further from any potential comparisons against prior work. Even for the two already-existing datasets with results included in the appendix, there are no references to previous SoTA.

2) Are the reported positive results meaningful?
Another problem with not showing results from previous work is that it is hard to understand whether the gains over the word2vec baseline are meaningful. For instance, the classification accuracy for the CNN corpus is increased from 92.05% (word2vec) to 92.29% (deGAN) and 92.36% (weGAN). The authors show that this is *statistically significant*, but is it meaningful? Is a 0.24% or 0.31% increase in accuracy worth the increased complexity of the model? For reference, what ratio of gains/computational cost would BERT incur?

3) What is the message of qualitative evaluation?
The authors offer qualitative analysis of their model. For instance, Table 2 shows the top 10 synonyms for words "Obama", "Trump" and "US", as produced by word2vec and weGAN. The lists of synonyms are extremely similar and don't seem to convey much more information besides the fact that weGAN makes minimal changes. I find the particular examples that the authors choose to underline the superiority of their algorithm unconvincing. For instance, the fact that, in the list of synonyms for US, the word British (selected by word2vec as the 9th closest word) is turned into American (on the same 9th position) loses its significance when you also consider the fact that weGAN did not derank any of the arguably less similar terms from higher positions (e.g. Turkish remains the 6th most similar term to US).

Other nit-picks:
- Formatting issues: In Sections 3.1 and 3.2, the numerical references to equations are broken. Figures 3 and 4 cover the text above. Tables 2, 4, 6, 7 are hard to read. It almost feels like the paper was updated from one conference format to another, without the necessary Latex adjustments.
- The paper has no "Conclusion" section
- Improper use of singular / plural of "corpus" / "corpora"

To end on a more positive note, here are some reasons why I did appreciate the paper:
- The usage of GANs in this context is very creative; the models do not generate *raw* text, but rather intermediate representations; this might be a promising direction for using GANs in NLP.
- The description of the models in sections 3.1 and 3.2 is rigorous, with useful explanations that break down the reasoning behind the loss function choices.

**Experience Assessment:**

I have read many papers in this area.

**Review Assessment: Checking Correctness Of Derivations And Theory:**

I assessed the sensibility of the derivations and theory.

**Review Assessment: Checking Correctness Of Experiments:**

I assessed the sensibility of the experiments.

**Review Assessment: Thoroughness In Paper Reading:**

I read the paper at least twice and used my best judgement in assessing the paper.

---

### Decision · Program_Chairs · 2019-12-19

**Decision:**

Reject

**Comment:**

The general consensus amongst the reviewers is that this paper is not quite ready for publication. The reviewers raised several issues with your paper, which I hope will help you as you work towards finding a home for this work.